# FINE-TUNING OFFLINE REINFORCEMENT LEARNING WITH MODEL-BASED POLICY OPTIMIZATION

## ABSTRACT

In offline reinforcement learning (RL), we attempt to learn a control policy from a fixed dataset of environment interactions. This setting has the potential benefit of allowing us to learn effective policies without needing to collect additional interactive data, which can be expensive or dangerous in real-world systems. However, traditional off-policy RL methods tend to perform poorly in this setting due to the distributional shift between the fixed data set and the learned policy. In particular, they tend to extrapolate optimistically and overestimate the action-values outside of the dataset distribution. Recently, two major avenues have been explored to address this issue. First, behavior-regularized methods that penalize actions that deviate from the demonstrated action distribution. Second, uncertainty-aware model-based (MB) methods that discourage state-actions where the dynamics are uncertain. In this work, we propose an algorithmic framework that consists of two stages. In the first stage, we train a policy using behavior-regularized model-free RL on the offline dataset. Then, a second stage where we fine-tune the policy using our novel Model-Based Behavior-Regularized Policy Optimization (MB2PO) algorithm. We demonstrate that for certain tasks and dataset distributions our conservative model-based fine-tuning can greatly increase performance and allow the agent to generalize and outperform the demonstrated behavior. We evaluate our method on a variety of the Gym-MuJoCo tasks in the D4RL benchmark and demonstrate that our method is competitive and in some cases superior to the state of the art for most of the evaluated tasks.

## 1 INTRODUCTION

Deep reinforcement learning has recently been able to achieve impressive results in a variety of video games (Badia et al., 2020) and board games (Schrittwieser et al., 2020). However, it has had limited success in complicated real-world tasks. In contrast, deep supervised learning algorithms have been achieving extraordinary success in scaling to difficult real-world datasets and tasks, especially in computer vision (Deng et al., 2009) and NLP (Rajpurkar et al., 2016). The success of supervised learning algorithms can be attributed to the combination of deep neural networks and methods that can effectively scale with large corpora of varied data. The previous successes of deep RL (Levine, 2016; Schrittwieser et al., 2020) seem to indicate that reinforcement learning can potentially scale with large active data exploration to solve specific tasks. However, the ability to collect such large datasets online seems infeasible in many real-world applications such as automated driving or robot-assisted surgery, due to the difficulty and inherent risks in collecting online exploratory data with an imperfect agent.

Existing off-policy RL algorithms can potentially leverage large, previously collected datasets, but they often struggle to learn effective policies without collecting their own online exploratory data (Agarwal et al., 2020). These failures are often attributed to the Q-function poorly extrapolating to out-of-distribution actions, which leads to overly optimistic agents that largely over-estimate the values of unseen actions. Because we train Q-functions using bootstrapping, these errors will often compound and lead to divergent Q-functions and unstable policy learning (Kumar et al., 2019).

Recently, there have been a variety of offline RL approaches that have attempted to address these issues. Broadly, we group these approaches into two main categories based on how they address the extrapolation issue.

The first set of approaches (Wu et al., 2019; Kumar et al., 2019) rely on behavior-regularization to limit the learned policy's divergence from the perceived behavioral policy that collected the data. These approaches discourage the agent from considering out-of-distribution actions in order to avoid erroneous extrapolation. While these methods can often be effective when given some amount of expert demonstrations, they often seem too conservative and rarely outperform the best demonstrated behavior.

The second set of approaches (Yu et al., 2020; Kidambi et al., 2020) leverage uncertainty-aware MB RL to learn a policy that is discouraged from taking state-action transitions where the learned model has low confidence. Thus, these methods allow a certain degree of extrapolation where the models are confident. Because these methods tend to be less restrictive, they can generalize better than behavior-regularization methods and sometimes outperform the behavioral dataset. However, this flexibility also seems to make it harder for these methods to recover the expert policy when it is present in the dataset, and reduce their effectiveness when trained with a narrow distribution.

In this work, we develop an algorithmic framework that combines ideas from behavior-regularization and uncertainty-aware model-based learning. Specifically, we first train a policy using behavior-regularized model-free RL. Then, we fine-tune our results with our novel algorithm Model-Based Behavior-Regularized Policy Optimization (MB2PO). We find that our approach is able to combine the upside of these approaches and achieve competitive or superior results on most of the Gym-MuJoCo (Todorov et al., 2012) tasks in the D4RL (Fu et al., 2020) benchmark.

## 2 RELATED WORK

While there exist many off-policy RL methods that can learn to solve a large variety of complex control tasks and can scale with large amounts of online data collection, these methods often perform quite poorly when run completely offline without any online data collection. Recently, there have been several methods that made progress in improving the capabilities of offline RL. For a general overview of the field of offline RL, we refer the reader to Levine et al. (2020). Here we will discuss some recent works that are particularly relevant to our approach.

### 2.1 IMPROVING OFF-POLICY Q-LEARNING

Many of the recent advances in both discrete and continuous action off-policy deep RL can be attributed to improvements in stabilizing off-policy Q-learning and reducing overestimation due to erroneous extrapolation. Some notable methods include target networks (Mnih et al., 2013), double Q-learning (DDQN) (van Hasselt et al., 2015), distributional RL (Bellemare et al., 2017; Dabney et al., 2017), and variance reduction through invertible transforms (Pohlen et al., 2018). In learning for continuous control, Fujimoto et al. (2018) introduced a conservative method that uses the minimum estimate of an ensemble of Q-networks as the target, which is often referred to as clipped double-Q-learning. Agarwal et al. (2020) demonstrated that Quantile Regression DDQN (Dabney et al., 2017) and other ensemble methods can be effective in certain discrete action offline RL problems. However, Agarwal et al. (2020) showed that when used naively, these methods do not perform well on complex continuous control tasks. In our work, we incorporate the mentioned advances in off-policy Q-learning into our approach to stabilize performance and prevent potential divergence.

Additionally, the offline RL algorithm Conservative Q-learning (CQL) (Kumar et al., 2020) has attempted to address Q-learning's overestimation issue on offline data directly by including a constraint term that discourages the agent from valuing an out-of-distribution action more than the demonstrated actions. In our method, instead of using a constraint on the Q-values, we use a combination of behavior-regularized model-free RL and uncertainty-aware model-based RL to discourage erroneous extrapolation.

### 2.2 BEHAVIOR-REGULARIZED MODEL-FREE RL

A variety of recent offline RL approaches have incorporated constraints or penalties on the learned policy's divergence from the empirical behavioral policy. In particular, recent works have used both KL Divergence (Wu et al., 2019) and mean measure of divergence (MMD) (Kumar et al., 2019).

MMD is sometimes used over KL Divergence because MMD approximately constrains the learned policy to be in the support of the behavioral policy, which is less restricting than KL Divergence. However, most behavior-regularization or policy-constraint methods require the behavioral policy to be represented explicitly in order to estimate these divergences or to enforce their policy constraint (Laroche et al., 2019). In contrast, AWAC (Nair et al., 2020) or CRR (Wang et al., 2020) is able to incorporate a KL divergence constraint without explicitly representing the behavioral policy. They do this by reformulating the policy-constrained RL optimization equations into a form that resembles behavioral cloning re-weighted by the exponential of the advantage. Wang et al. (2020) demonstrates that this method can effectively learn complex control tasks purely from offline data, and Nair et al. (2020) demonstrate that performance can even be improved with further online data collection. In this work, we demonstrate that these properties make AWAC work exceptionally well when used for initialization as well as when used for fine-tuning with Model-Based Policy Optimization (MBPO) (Janner et al., 2019).

### 2.3 UNCERTAINTY-AWARE MODEL-BASED RL

MB RL algorithms have several natural advantages for offline RL compared to model-free RL algorithms. First, MB RL algorithms rely on supervised learning, which provide more robust gradient signals compared to bootstrapped learning and policy gradients. Second, learning a dynamics model often provides strong task-independent supervision, which allows MB RL algorithms to learn from sub-optimal trajectories. These benefits make generalization easier, and can allow MB RL algorithms to surpass the performance of the demonstrated data. In fact, in many environments, MB RL methods have already been effective in learning with offline or randomly collected datasets. Additionally, there is a rich history of prior works that have explored robust solutions to MDPs with uncertain transition dynamics (Nilim & El Ghaoui, 2005; Iyengar, 2005). However, it can be difficult to scale these type of methods to high-dimensional continuous control tasks, especially when using deep neural networks. Recently, incorporating uncertainty estimation techniques from supervised learning in MB RL has demonstrated further improvement in both online (Chua et al., 2018) and offline deep RL. In particular, two recent works, Model-Based Offline Policy Optimization (MOPO) (Yu et al., 2020) and Model-Based Offline Reinforcement Learning (MoREL) (Kidambi et al., 2020), have demonstrated impressive results by incorporating uncertainty-aware MB RL with the Dyna (Sutton, 1991) style algorithm MBPO (Janner et al., 2019). Both methods use these models to create conservative MDPs that have a lower potential expected sum of rewards compared to the true MDP. By performing policy optimization in the conservative MDP through MBPO they are able to learn a conservative policy that can outperform the demonstrated trajectories. However, these methods can often fail to recover the expert policy even though it was demonstrated in the dataset. We believe that this is largely due to a lack of effective methods for estimating epistemic uncertainty for neural network regression.

## 3 PRELIMINARIES

In RL, we assume our agent operates within a standard Markov decision process (MDP) $M = (\mathcal{S}, \mathcal{A}, T, r, \rho_0, \gamma)$, where $\mathcal{S}$ denotes the state space, $\mathcal{A}$ denotes the action space, $T(s'|s, a)$ represents the probabilistic transition dynamics, $r$ is the reward function, $\rho_0$ is the initial state distribution, and $\gamma \in (0, 1)$ is the discount factor. The objective in RL is to learn a policy $\pi(a|s)$ that optimizes the expected discounted sum of rewards $R^\pi = \mathbb{E}_{\pi, T, \rho_0}[\sum_{t=0}^\infty \gamma^t r(s_t, a_t)]$.

In offline RL, we assume that during training we only have access to a fixed dataset $\mathcal{D}_\beta$ containing a set of tuples $(s, a, s', r)$ of environment transitions and associated rewards. We assume that the data was collected by a policy $\pi_\beta$, which we call the behavioral policy. Typically, when training with data not collected by your current policy $\pi$, we either use off-policy model-free algorithms or model-based algorithms. The most common off-policy model-free algorithms are actor-critic algorithms that use policy iteration. Policy iteration involves alternating between policy evaluation and policy improvement in order to learn an effective policy. In policy evaluation, these methods train a parametric Q-function by iteratively minimizing the temporal difference equation

$$Q_{k+1}^\pi = \arg\min_Q \mathbb{E}_{s,a,s'\sim\mathcal{D}} \left[ ((r(s,a) + \gamma \mathbb{E}_{a'\sim\pi(\cdot|s')}[Q_k^\pi(s', a')]) - Q^\pi(s, a))^2 \right] \qquad (1)$$

In policy improvement, we update our parametric policy $\pi$ to maximize our current Q-function

$$\pi_{k+1} = \arg\max_{\pi} \mathbb{E}_{s \sim \mathcal{D}, a \sim \pi(\cdot|s)}[Q^{\pi_k}(s,a)] \tag{2}$$

In MB RL, we attempt to learn a model $\hat{T}$ of the transition dynamics and a model $\hat{r}$ of the reward function. With this learned model of the dynamics and reward function we can create a model MDP $\hat{M} = (\mathcal{S}, \mathcal{A}, \hat{T}, \hat{r}, \rho_0, \gamma)$ to estimate the true underlying MDP $M$. These methods tend to use either trajectory optimization or policy optimization in the model MDP to produce their policy.

## 4 MODEL-BASED BEHAVIOR-REGULARIZED POLICY OPTIMIZATION FOR OFFLINE FINE-TUNING

For many offline datasets, it could be much harder to learn an effective model of the MDP than to learn a reasonable policy. This is especially the case when there is low variability or insufficient coverage of the state and action space in the collected dataset, or in environments with complex observations, like images, or long horizons. To overcome these issues, recent works (Yu et al., 2020; Kidambi et al., 2020) have leveraged uncertainty estimation methods in order to construct conservative MDPs that use soft penalties or hard thresholds on model uncertainty to discourage deviating from the confident regions. However, these methods rely on the efficacy of ensemble-based neural network uncertainty estimation methods which currently are not particularly effective at estimating epistemic uncertainty in regression settings. Therefore, we propose Model-Based Behavior-Regularized Policy Optimization (MB2PO). In MB2PO, we likewise use uncertainty-aware models to perform offline MBPO, but use the behavior-regularized model-free algorithm AWAC (also known as CRR-exp) instead of SAC (Haarnoja et al., 2018) for policy optimization.

### 4.1 CONSERVATIVE MBPO

In this work, we use MOPO (Yu et al., 2020) as a basis for our conservative MBPO, due to its simplicity and prior effective results on the D4RL benchmarks. In MOPO, they construct a conservative MDP by augmenting the reward function as follows

$$\tilde{r}(s,a) = \hat{r}(s,a) - \lambda u(s,a) \tag{3}$$

where $\hat{r}$ is the learned estimate of the reward and $u$ is the estimated uncertainty for the model transition. Note, that this general formulation for a conservative MDP has also been explored in other prior work such as (Ghavamzadeh et al., 2016). Still, we specifically follow MOPO in using the maximum standard deviation across an ensemble of probabilistic dynamics models as our measure of uncertainty. Therefore, we can decompose our Q-function in this conservative MDP as

$$Q^{\pi}(s,a) = \hat{Q}_r^{\pi}(s,a) - \lambda Q_u^{\pi}(s,a) \tag{4}$$

where $\hat{Q}_r^{\pi}$ represents our estimate of the expected discounted sum of rewards in the real MDP and $Q_u^{\pi}$ represents our expected discounted sum of uncertainty penalties. Now at convergence, if our policy $\pi$ deviates from the behavioral policy $\pi_{\beta}$ that collected the data, then we expect for all states in the conservative MDP that

$$\mathbb{E}[Q^{\pi}] \geq \mathbb{E}[Q^{\pi_{\beta}}] \tag{5}$$

Thus, by plugging in our decomposition we get

$$\mathbb{E}[\hat{Q}_r^{\pi}(s,a)] \geq \mathbb{E}[\hat{Q}_r^{\pi_{\beta}}(s,a)] + \lambda \mathbb{E}[Q_u^{\pi}(s,a)] \tag{6}$$

While in theory, with well-calibrated uncertainty estimates and a proper tuning of $\lambda$, this should lead to only safe policy improvements over the behavioral policy, in practice it seems that MOPO is often unable to recover expert-level performance when it is provided in the offline dataset. This is unsurprising given that it is hard to generate well-calibrated epistemic uncertainty estimates in regression settings, and there will inevitably be model errors that will lead to overestimated Q-values.

To address these issues, we use policy constrained model-free RL in MB2PO. In policy constrained model-free RL, we attempt to optimize the following policy objective

$$\pi = \arg\max_{\pi} \mathbb{E}_{a \sim \pi(\cdot|s)}[Q^{\pi}(s,a)] \tag{7}$$

$$\text{s.t. } D_{\mathrm{KL}}(\pi(\cdot|s)\|\pi_{\beta}(\cdot|s)) \leq \epsilon$$

If we estimate both $\pi$ and $\pi_\beta$ to be roughly univariate Gaussians with similar variances, then the KL constraint becomes an $\ell_2$ constraint on the policy mean. Because we expect our models to be locally accurate around the data, this constraint can help ensure that we stay in the effective region of the estimated MDP even if we have poorly calibrated uncertainty estimation. Additionally, Janner et al. (2019) demonstrated that the difference between the true expected returns $J(\pi)$ and the expected returns $\hat{J}(\pi)$ of an MDP induced by an approximate model can be bounded by

$$J(\pi) \geq \hat{J}(\pi) - \left[ \frac{2\gamma r_{\max}(\epsilon_m + 2\epsilon_\pi)}{(1-\gamma)^2} + \frac{4r_{\max}\epsilon_\pi}{1-\gamma} \right] \tag{8}$$

where $r_{\max}$ is the maximum reward, $\gamma$ is the discount factor, $\epsilon_m$ is a bound on the total variation distance (TVD) between the learned model and the true model, and $\epsilon_\pi$ is a bound on the TVD between $\pi$ and $\pi_\beta$ on the demonstrated states. By Pinker's inequality, bounding the KL divergence also bounds the TVD. Therefore, by leveraging policy constraints in the policy optimization in MBPO, we can reduce the gap in expected returns and improve the algorithm's robustness to model errors.

## 4.2 BEHAVIOR-REGULARIZED MODEL-FREE RL WITH AWAC

For performing behavior-regularized policy optimization, we use AWAC (Nair et al., 2020), also known as CRR-exp (Wang et al., 2020) due to its impressive results in offline RL and its ability to be fine-tuned with additional online data.

By enforcing the KKT conditions (Peng et al., 2019; Peters & Schaal, 2007; Gómez et al., 2014), we can derive an analytic solution to Equation 7, where the Lagrangian is

$$\mathcal{L}(\pi, \alpha) = \mathbb{E}_{a \sim \pi(\cdot|s)}[Q^\pi(s,a)] + \alpha(\epsilon - D_{\mathrm{KL}}(\pi(\cdot|s)\|\pi_\beta(\cdot|s)))$$

We can substitute $A^\pi(s,a)$ for $Q^\pi(s,a)$ because it does not affect the optimum and get the closed-form solution

$$\pi^*(a|s) = \frac{1}{Z(s)}\pi_\beta(a|s)\exp\left(\frac{A^\pi(s,a)}{\alpha}\right)$$

where $Z(s)$ is the normalizing partition function. In order to project this solution into our policy space, we update our parameters by minimizing $D_{\mathrm{KL}}(\pi^*\|\pi_\theta)$. This leads to the following iterative update

$$\theta_{k+1} = \arg\min_\theta \mathbb{E}_{s,a \sim D}\left[-\log \pi_\theta(a|s)\frac{1}{Z(s)}\exp\left(\frac{A^{\pi_k}(s,a)}{\alpha}\right)\right] \tag{9}$$

We follow Wang et al. (2020) and Peng et al. (2019) and avoid estimating $Z(s)$ and instead clamp the exponential term to be at most 20. Additionally, one could adaptively learn $\alpha$ using dual gradient descent, but this would require us to explicitly model the behavioral policy $\pi_\beta$. Instead, we use a fixed $\alpha$ for all of our results. Additionally, the Q-function is updated off-policy using the Bellman equations as described in Equation 2 and the improvements from section 2.1.

One of the major benefits of using AWAC with a fixed $\alpha$ is that we can leverage behavior regularization in a principled manner without needing to explicitly represent the behavioral policy. This is particularly important in 3 major cases: 1. when there are not enough data to learn the behavioral policy; 2. when the data was collected by a variety of different policies or sources; 3. when the data was collected by a policy outside of your policy class, such as a human expert or a controller that leverages hidden state information.

Additionally, we can view AWAC as a reweighted behavioral cloning algorithm. Unlike SAC (Haarnoja et al., 2018) and DDPG (Lillicrap et al., 2015), it does not rely on the reparametrization trick or gradients of your learned Q-function to perform policy updates. This allows us to use a wider ranger of policy classes, which in this work we take advantage of by using a tanh squashed GMM with 5 components. We suspect that there are also some additional benefits to not depending on the gradients of the learned Q-function, which might be particularly bad in offline settings, but leave further investigation to future work.

An important thing to note with AWAC is that we can influence the implicit behavioral penalty by controlling the source of the data we train with. This holds, for example, if we perform a series

of policy updates only using data collected by the previous policy iterate. Then, we are implicitly performing a trust-region policy update like TRPO (Schulman et al., 2015) and PPO (Schulman et al., 2017) of the form

$$\pi_{k+1} = \arg\max_{\pi} \mathbb{E}_{a\sim\pi(\cdot|s)}[Q^{\pi_k}(s,a)] \tag{10}$$
$$\text{s.t. } D_{\mathrm{KL}}(\pi(\cdot|s)\|\pi_k(\cdot|s)) \leq \epsilon$$

In fact, if we train on data collected by the last $n$ policy iterates, then we are approximately constraining our policy to a weighted sum of the previous $n$ policies $\pi_k^{(n)} = \frac{1}{n}\sum_{i=0}^{n-1}\pi_{k-i}$ and damping our learning process in the policy space.

In our work, we train with a $\omega \in [0,1]$ portion of the data from offline data collected by $\pi_\beta$ and a $(1-\omega)$ portion of the data collected online from the last $n$ policy iterates in the conservative MDP defined by our learned models. Therefore, we are approximately optimizing the following objective

$$\mathbb{E}_{a\sim\pi(\cdot|s)}\left[\hat{Q}^\pi(s,a)\right] - \alpha\left(\omega D_{\mathrm{KL}}(\pi(\cdot|s)\|\pi_\beta(\cdot|s)) + (1-\omega)D_{\mathrm{KL}}(\pi(\cdot|s)\|\pi_k^{(n)})\right) \tag{11}$$

Therefore, by using AWAC as the policy optimization algorithm in MB2PO, we can easily perform behavior-regularized policy optimization with soft damped trust region updates in the conservative MDP to reduce the effects of model errors and poor uncertainty estimation.

### 4.3 MODEL-BASED BEHAVIOR-REGULARIZED POLICY OPTIMIZATION

---

Train $\pi_\theta$, $Q_\phi$ with AWAC with samples from $\mathcal{D}_\beta$
Train an ensemble of N probabilistic dynamics
$\{\hat{T}_\theta^i(s_{t+1},r|s_t,a_t) = \mathcal{N}(\mu_\theta^i(s_t,a_t), \Sigma_\theta^i(s_t,a_t))\}_{i=1}^N$ on the data in $\mathcal{D}_\beta$
**for** *epoch k= 1, 2, ...* **do**
    Initialize empty replay buffer $\mathcal{D}_k$
    **for** *1, 2, ..., batchsize* **do**
        Sample state $s_1$ from $\mathcal{D}_\beta$
        **for** *j = 1, 2, ..., h* **do**
            $a_j \sim \pi(s_j)$
            Uniformly sample $\hat{T}$ from $\{\hat{T}^i\}_{i=1}^N$
            $s_{j+1}, r_j \sim \hat{T}(s_j, a_j)$
            $\tilde{r}_j = r_j - \lambda\max_{i=1}^N\|\Sigma^i(s_j, a_j)\|_F$
            Add sample $(s_j, a_j, \tilde{r}_j, s_{j=1})$ to $\mathcal{D}_k$
        **end**
    **end**
    Draw $\omega$ portion of the samples from $D_\beta$ and the rest uniformly from $\{\mathcal{D}_{k-i}\}_{i=0}^{99}$ to train $\pi_\theta$
      and $Q_\phi$ with AWAC
**end**

---

We first initialize our policy by training with AWAC solely on the offline data.

Next for fine-tuning with MB2PO, we train an ensemble of probabilistic dynamics models represented by neural networks that output a diagonal Gaussian distribution over the next state and reward: $\{\hat{T}_\theta^i(s_{t+1},r|s_t,a_t) = \mathcal{N}(\mu_\theta^i(s_t,a_t), \Sigma_\theta^i(s_t,a_t))\}_{i=1}^N$. We construct a conservative MDP that at every time step uses a randomly drawn dynamics model from $\{\hat{T}_\theta^i\}_{i=1}^M$ to determine the next state transition. Additionally, we incorporate an penalty on the largest predicted standard deviation among the dynamics models as a practical means of penalizing both epistemic and aleatoric uncertainty.

Then, we alternate between collecting data with our current policy in the conservative MDP and updating our policy and Q-network using Equation 9 and Equation 2 respectively. When collecting data in the conservative MDP, we collect $h$-length truncated trajectories starting from states in the original offline dataset. By collecting data this way, we are able to collect a variety of imagined

data without relying on long model rollouts, which would inevitably lead to compounding errors. When performing training updates, we sample $\omega \in [0, 1]$ of the data from the original dataset and the remaining $1 - \omega$ uniformly from the last 100 policy iterates. Our full algorithm is outlined in Algorithm 1.

## 5 EXPERIMENTS

| Task and Dataset | AWAC | AWAC + MB2PO(Ours) | MOPO | BEAR | BRAC-v | CQL ($\mathcal{H}$) |
|---|---|---|---|---|---|---|
| halfcheetah-random | 18.8 ±1.5 | 25.5 ±1.1 | 31.9 ± 2.8 | 25.5 | 28.1 | **35.4** |
| hopper-random | 11.2 ±0.1 | 11.4 ±0.1 | **13.3** ± 1.6 | 9.5 | 12.0 | 10.8 |
| walker2d-random | 1.4 ±3.0 | 0.2 ±2.3 | **13.0** ± 2.6 | 6.7 | 0.5 | 7.0 |
| halfcheetah-medium | 40.9 ±0.3 | 40.7 ±0.2 | 40.2 ± 2.7 | 38.6 | **45.5** | 44.4 |
| hopper-medium | 35.0 ±3.9 | 55.7 ±14.6 | 26.5 ± 3.7 | 47.6 | 32.3 | **58.0** |
| walker2d-medium | 74.3 ±1.6 | 80.4 ±0.9 | 14.0 ± 10.1 | 33.2 | **81.3** | 79.2 |
| halfcheetah-expert | 106.7 ±0.6 | 105.1 ±1.3 | | **108.2** | -1.1 | 104.8 |
| hopper-expert | 108.1 ±3.8 | 105.5 ±10.5 | | **110.3** | 3.7 | 109.9 |
| walker2d-expert | 100.5 ±12.3 | 107.4 ±1.1 | | 106.1 | 0.0 | **153.9** |
| halfcheetah-medium-expert | 104.7 ±1.6 | **104.8** ±1.1 | 57.9 ± 24.8 | 51.7 | 45.3 | 62.4 |
| hopper-medium-expert | 75.1 ±15.9 | 79.1 ±13.5 | 51.7 ± 42.9 | 4.0 | 0.8 | **111.0** |
| walker2d-medium-expert | 81.8 ±18.9 | 86.2 ±35.5 | 55.0 ± 19.1 | 26.0 | 66.6 | **98.7** |
| halfcheetah-mixed | 42.3 ±0.3 | **55.6** ±0.6 | 54.0 ±2.6 | 36.2 | 45.9 | 46.2 |
| hopper-mixed | 30.1 ±0.9 | 72.6 ±25.5 | **92.5** ±6.3 | 25.3 | 0.8 | 48.6 |
| walker2d-mixed | 16.9 ±1.7 | **61.9** ±8.2 | 42.7 ±8.3 | 10.8 | 0.9 | 26.7 |

Table 1: Here we compare AWAC (averaged over 4 seeds) and AWAC + MB2PO (averaged over 4 seeds) to recent offline model-free and model-based RL algorithms. We report the normalized score where 100 is the performance of a fully trained SAC policy and 0 is the performance of a uniform random policy. For the other methods, we report the results from their own papers or the original D4RL paper. "-expert" results for MOPO were not included in the original paper and thus are omitted here. We include the stand deviation for our results and for previous results if reported. We bold the highest mean.

In our experiments, we aim to address two questions: (1) Is AWAC an effective initialization algorithm? (2) Can we further improve performance by fine-tuning with MB2PO?

We evaluate (1) by comparing AWAC to other state-of-the-art model-free offline RL algorithms. In particular, we compare our results to BRAC-v (Wu et al., 2019), BEAR (Kumar et al., 2019), and CQL (Kumar et al., 2020) on the Gym-MuJoCo tasks in the D4RL benchmark.

We evaluate (2) by fine-tuning the policy and Q-function, after running AWAC for 500000 gradient steps, with MB2PO. In addition to the model-free offline RL algoirthm above, we also compare these results to MOPO, which to the best of our knowledge is the state-of-the-art MB offline RL algorithm on the Gym-MuJoCo tasks in the D4RL benchmark. All of our hyperparameters for AWAC and MB2PO are given in the appendix.

The Gym-MuJoCo tasks are a standard in evaluating modern deep RL algorithms. The goal in these tasks is to learn to travel as far forward as possible within a set horizon on a variety of different robots. The D4RL benchmark contains a variety of precollected datasets for the halfcheetah, walker2d, and hopper tasks. For each robot task, there are 5 different provided datasets. The "-random" datasets contain 1 million samples collected from a randomly initialized policy. The "-medium" datasets contain 1 million samples collected from a RL policy partially trained to a performance of approximately 33. The "-expert" datasets contain 1 million samples collected from a fully trained RL policy that reaches approximately 100. The "-mixed" datasets contain all the data in the replay buffer from the partially trained "medium" policy. Finally, the "-medium-expert" datasets are a combination of the "-medium" and "-expert" datasets. An important thing to note is that all datasets besides the "-mixed" datasets were collected with only 1 or 2 policies, and thus probably only cover a narrow part of the state-action distribution. On the other hand, the "-mixed" dataset contains the data collected by all of the policy iterates during an incomplete RL training run, and thus represents a much wider part of the state-action distribution.

Results in Table 1 demonstrate that AWAC on its own can get reasonable results on all the datasets and can approach state-of-the-art results on "-expert" and "-medium-expert" datasets. Unlike the other behavior-regularized model-free methods, AWAC and CQL are able to get near or fully recover expert-level performance when trained on the "medium-expert-" datasets. This indicates that AWAC and CQL are more robust as there is less of a drop in performance compared to other methods when incorporating additional sub-optimal trajectories.

Next, we fine-tune the trained AWAC policies with MB2PO. For each task and dataset, we pretrain an ensemble of 5 probabilistic dynamics models for 100000 gradient steps on the behavioral dataset. We then perform MB2PO for 500 iterations. Each iteration consists of collecting 1000000 steps from $h$-length truncated trajectories in the conservative MDP, which should run in a few seconds on modern GPU hardware, followed by 1000 gradient steps.

Results in Table 1 demonstrate that our method is effective in improving the performance over AWAC in 11 of the 15 tasks. In particular, we find that MB2PO significantly improves the performance on all of the "-mixed" datasets and even achieves state-of-the-art on "walker2d-mixed" by a large margin. These strong results in the "-mixed" datasets demonstrate that our model-based fine-tuning method can be especially beneficial when there is sufficient variation in the behavioral dataset. Additionally, the noticeable improvement in some of the "-medium" and "-medium-expert" datasets demonstrate that our fine-tuning can be effective even when the data was collected by one or two policies. In the 4 cases where MB2PO fine-tuning degrades performance, it is always negligible and never over 3 points.

Our method also outperforms MOPO, the most direct comparison, in 8 out of the 12 comparable tasks. These results demonstrate the benefits of combining behavior-regularized model-free RL with uncertainty-aware MB RL as we are able to get the best of both worlds. We are able to recover high-level performance when available in the dataset like AWAC, and we can still generalize and learn to outperform the best observed trajectory like MOPO.

Finally, our method is quite competitive with CQL as we beat it for 7 of the 15 tasks, and generally our results are quite comparable. However, our method significantly outperforms CQL in all of the "-mixed" datasets. These results indicate that our method might be superior in situations where the dataset was collected by a variety of different policies.

## 6 CONCLUSION

We proposed an algorithmic framework that leverages the benefits of both behavior-regularized model-free methods and uncertainty-aware model-based methods. We do this by first training an initial policy with the offline model-free AWAC algorithm. Then, we fine-tune with our novel MB2PO algorithm. We perform this by learning uncertainty-aware models that are used to create a conservative MDP. Then, we continue to use AWAC to further update our policy and Q-function in this conservative MDP. By using AWAC, we are able to perform policy optimization while implicitly constraining the learned policy's KL divergence to the behavioral policy. We demonstrate that this two-stage process allows us to get the best of both worlds between behavior-regularized model-free methods and uncertainty-aware model-based methods. Specifically, the initial AWAC training allows us to often recover the best-performing behavior in the dataset, and, when possible, MB2PO fine-tuning can allow us to generalize and outperform the demonstrated behavior.

We see four important directions of future work in order to extend the effectiveness and applicability of MB2PO: 1. developing a rigorous means of determining for what datasets MB2PO fine-tuning can be effective; 2. improving MB RL and neural network uncertain estimation to increase the number of datasets where MB2PO can be effective; 3. better leveraging behavior-regularization in the policy optimization or the conservative MDP to improve MB2PO's ability to recover expert behavior when available; 4. improving off-policy evaluation (Thomas et al., 2015) for neural network policies in order to facilitate offline hyperparameter tuning.

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

# 7 APPENDIX

All methods were trained with the Adam optimizer(Kingma & Ba, 2014)

## 7.1 AWAC

Our Q-networks and policies were represented with [256, 256, 256, 256] fully connected networks with relu hidden-activations. The policy was a 5-head tanh squashed GMM. The Q-network out-putted 100 quantiles and was trained with the following Q-learning improvement: Quantile Regression DQN (Dabney et al., 2017), Clipped DQN(Fujimoto et al., 2018), Double DQN(van Hasselt et al., 2015), and Invertible Transforms(Pohlen et al., 2018). The advantage was estimated with 10 samples. We ran AWAC for 500000 gradient steps. The rest of the parameters were taken from the original AWAC paper (Nair et al., 2020): $\alpha = 1.$, batch size 1000, policy weight decay $1.e - 4$, policy and Q learning rate $3.e - 4$, soft target network update $5.e - 3$.

## 7.2 MB RL

Our probabilistic dynamics models were represented with [200, 200, 200, 200] fully connected networks with swish hidden-activations. The models outputted a mean and variance for every state variable and the reward. We trained the models for 100000 gradient steps. The rest of the parameters are: batch size 256, weight decay $1.e - 4$, learning rate $1.e - 3$.

## 7.3 MB2PO

MB2PO used the same AWAC parameters as before, but changed $\alpha$ according to the table below. We ran MB2PO for 500 iterations where each iteration consisted of collecting 1000000 samples from truncated $h$-length trajectories with the current GMM policy, then training for 1000 steps with $\omega$ of the data from real dataset and $1 - \omega$ of the data from imagined model rollouts from the last 100 iterations. We picked the better of $\alpha = 1, \omega = \{0.01, 0.05\}$ for the mixed datasets, and the better of $\alpha = \{1, 2\}, \omega = 0.8$ for the other datasets. We used $h = 5$ for several of the halfcheetah datasets and $h = 1$ for the rest because any larger $h$ tended to cause the Q-values to explode.

| Task and Dataset | alpha | omega | h | lambda |
|---|---|---|---|---|
| halfcheetah-random | 1. | 0.8 | 5 | 1 |
| hopper-random | 1. | 0.8 | 1 | 1 |
| walker2d-random | 1. | 0.8 | 1 | 1 |
| halfcheetah-medium | 1. | 0.8 | 5 | 1 |
| hopper-medium | 1. | 0.8 | 1 | 1 |
| walker2d-medium | 1. | 0.8 | 1 | 1 |
| halfcheetah-expert | 2. | 0.8 | 1 | 1 |
| hopper-expert | 2. | 0.8 | 1 | 1 |
| walker2d-expert | 2. | 0.8 | 1 | 1 |
| halfcheetah-medium-expert | 2. | 0.8 | 1 | 1 |
| hopper-medium-expert | 2. | 0.8 | 1 | 1 |
| walker2d-medium-expert | 2. | 0.8 | 1 | 1 |
| halfcheetah-mixed | 1. | 0.01 | 5 | 1 |
| hopper-mixed | 1. | 0.01 | 1 | 1 |
| walker2d-mixed | 1. | 0.01 | 1 | 1 |

