# OpenReview forum: "Fine-Tuning Offline Reinforcement Learning with Model-Based Policy Optimization"
_ICLR.cc/2021/Conference — Reject_

### Official Review · AnonReviewer4 · 2020-10-26
**Official Blind Review #3**

**Rating:** 3
**Confidence:** 4

**Review:**

The authors take an attempt at offline RL thanks to a mix between behavioural policy regularization and model based policy optimization. They basically combine two algorithms: AWAX and, depending on the level of safety given an epistemic uncertainty evaluation, MOPO may be additionally used to fine tune the policy.

Unfortunately the work suffers from several severe weaknesses:
- the writing is not good. See the typo and minor comments section.
- the positioning is biased and missing to many accounts. Out of 31 citations, 12 are from the same author. Even more problematic, most, if not all, the references on offline RL are from this author, and therefore lacks diversity. In particular Model-based offline RL [Iyengar2005,Nilim2005], and model-free offline RL [Thomas2015b] have a rich history. More specifically, Equation (3) is identical to that of the Reward-Adjusted MDP found in [Petrik2016]. The Safe Policy Improvement objective has been considered for instance in [Thomas2015b,Petrik2016]. Equation (7) proposes to optimize the policy under a constraint on the policy search (identical to online TRPO, which is evoked later) that is very similar to [Laroche2019], except that the constraint is not state based and therefore probably less efficient.
- "If our initial policy does not achieve expert level performance, and we are confident that we can learn an effective model with the available data, then ..." => it is unclear how these decisions are sorted out. Performing those safety tests are an area of research in themselves [Thomas2015a].
- even if we assume that the algorithmic novelty is proven, it seems pretty incremental, since it amounts to perform a test to decide between two algorithms.
- Finally, the experimental results do not savethe day. We observe that "Ours" is always the max of AWAC and AWAC+MB2PO, which is a little suspicious, since we have no information on how the decision is made. In comparison with CQL, it is not better (but it is a strong baseline). So, it's not improving the state of the art. It would have been informative to show the behavioural performance in each setting.

Typo and minor comments:
- AWAC is used without citation or explanation first (2.1)
- "The most common off-policy model-free algorithms are actor-critic algorithms that alternate between policy evaluation and policy improvement in order to learn an effective policy." => this is not actor-critic but policy iteration.
- "Otherwise, we use the fully trained AWAC policy. These results are reported in the column Ours in Table 1." => otherwise what?
- Sec. 4.2: effect => affect
- Sec. 4.3: degredation => degradation

[Iyengar2005] Iyengar, G. N. Robust dynamic programming. Mathematics of Operations Research, 30(2):257–280, 2005.
[Laroche2019] Laroche, R., Trichelair, P., & Tachet des Combes, R. T. (2019, May). Safe policy improvement with baseline bootstrapping. In International Conference on Machine Learning (pp. 3652-3661).
[Nilim2005] Nilim, A. and El Ghaoui, L. Robust control of Markov decision processes with uncertain transition matrices. Operations Research, 53(5):780–798, 2005.
[Petrik2016] Petrik, M., Ghavamzadeh, M., & Chow, Y. (2016). Safe policy improvement by minimizing robust baseline regret. In Advances in Neural Information Processing Systems (pp. 2298-2306).
[Thomas2015a] Thomas, P. S., Theocharous, G., & Ghavamzadeh, M. (2015, February). High-confidence off-policy evaluation. In Twenty-Ninth AAAI Conference on Artificial Intelligence.
[Thomas2015b] Thomas, P., Theocharous, G., & Ghavamzadeh, M. (2015, June). High confidence policy improvement. In International Conference on Machine Learning (pp. 2380-2388).

---

> ### Author Response · Authors · 2020-11-25
> **Addressing Concerns**
>
> * We believe we addressed the typos in the most recent revision
>
> * First, we want to say thank you for pointing us to these works. We have included citations to most of them in our most recent revision.
>
> * In our most recent revision, we were able to improve our results to the point where this check is no longer necessary. We were able to improve the results such that now we can always get good results with fine-tuning with MB2PO. In fact, it improves the results over AWAC on 11 of the 15 tasks, and it only causes a negligible drop (< 3 points max) in the other 4 cases.
>
> * Hopefully, our new results address this concern. As our method MB2PO is able to outperform AWAC in 11 of the 15 tasks and MOPO in 8 of the 12 comparable tasks.
>
> * Again, hopefully our new results address this concern. While admittedly CQL still beats us in 8 of the 15 tasks. We significantly outperform CQL in all the "-mixed" datasets, which indicates that our method might be better for datasets collected by a variety of policies. We believe that this is an important distinction because many potential use cases for offline RL will probably involve data collected from a variety of sources.

---

### Official Review · AnonReviewer3 · 2020-10-28
**potentially interesting idea, good results but contribution not clearly highlighted**

**Rating:** 5
**Confidence:** 4

**Review:**

**Final recommendation**
I do not recommend accepting the paper. The results have been greatly improved. They now look decent and I have improved my score as a result. However I think the contributions are still not clearly highlighted. I however encourage the authors to improve their paper.

**Summary**
This paper proposes to combine two approaches for offline reinforcement learning: behavior-regularized methods and uncertainty-aware model-based methods. The proposed approach works in two steps. First, a conservative MDP is constructed by estimating a transition functin and substracting an uncertainty penalty from the reward for each state and action pair, similarly to MOPO. (Yu et al., 2020).  AWAC (Nair et al., 2020) is used to learn a policy $\pi$ using offline data and the penalized reward, where $\pi$ is constrained to be closed to the behavior policy in terms of KL divergence. In a second, optional step called MB2PO, the paper proposes to iteratively refine $\pi$ by sampling trajectories from the conservative MDP and improving $\pi$ using AWAC. The proposed approach is compared to existing methods (MOPO, BEAR, BRAC, CQL) on the D4RL benchmark. The best result of the proposed approach, i.e. with or without the second step, is shown to be competitive with the state of the art.

---
**Strong points**

1) The paper offers a possible theoretical reason about why the proposed approach may improve result: it improves the bound on the difference between the return of the MDP and the approximate MDP.

2) The experiments look good to me and show the performance of the method.

3) Results can be competitive with the state of the art.

**Weak points**

1) My key concern about the paper is that, at the moment, it does not provide a (non empirical) method to decide whether to used the second step or not. In the experiments the best out of the two results is used. This makes it hard to assess the interest of the method.

2) Reading the paper was rather easy, but identifying the difference between the contributions of the paper and previous work was not so easy for me.

3) I am not sure I could reproduce the experiments. For example the architecture of the models used is not described.

---
**Recommendation**

I vote for rejecting the paper. The idea looks interesting and should be investigated further. However, currently, the proposed idea is in my opinion not fully developped (weak point 1) and the results, while fine, are not improving the state of the art enough to overlook this point.

---
**Details**

I like the theoretical intuition. While not the foundation of the idea, it is good to have a theoretical sanity check.

The second step of the method sometimes severely deteriorates the policy. For me, evaluating both policies and picking the best one is not a very satisfactory method for an off-policy algorithm.

Regarding the contributions of the paper, I am under the impression that in the section presenting the method (section 4), the novel contributions are Equation 7 (and associated text) in section 4.1, Equation 11 (and associated text) and section 4.3. It is however nice to have information about existing methods leveraged in this work. I am not sure what the best solution is but I would like to suggest using "they" rather than "we" when the paper describes previous work.

Around Equation 6, the paper states that a problem with MOPO is that estimating uncertainty is difficult and *there will inevitably be model errors that will lead to overestimated Qvalues* and that the proposed approach address this issue. I think it would be nice to have some experimental results highlighting that the proposed approach indeed fixes this issue, perhaps by showing the uncertainty estimates and different resulting actions for different methods in problematic state.


---
**Questions**

Could you please correct me if I misunderstood the contributions of the paper?

---
**Minor details**
Section 2.1: I think *AWAC* is used before being introduced.
Algorithm 1 has no caption

---

> ### Author Response · Authors · 2020-11-25
> **Addressing Concerns**
>
> Weak Points
> 1. In our most recent revision, we were able to improve the results such that now we can always get good results with fine-tuning with MB2PO. In fact, it improves the results over AWAC on 11 of the 15 tasks, and it only causes a negligible drop (< 3 points max) in the over 4 cases.
>
> 2. We believe that the main benefit of this work is in proposing a method that leverages both behavior-regularized model-free RL and uncertainty-aware MB RL for offline RL, and in demonstrating both theoretically and empirically the merits of such a combined approach.
>
> 3. We have included a lot of the hyperparameters we could think of in the appendix. Also, we would be happy to share our code once things are deanonymized.
>
> Details
>
> 2. With our improved results, hopefully this is not a concern.
>
> 3. (in the original version) equation 7 is not novel, but we believe the accompanying text in 4.1 is. equation 11 and associated text is novel. Most of 4.3  follows from the original MOPO paper. For 4.3, the main novelty is in using AWAC for the policy optimization and keeping track of the data from just the last 100 policy iterations
>
> 4. While I do not have explicit experimentally results, I can recount that the Q-function would often explode if alpha was less than 0.5 for the not "-mixed" datasets.

---

### Official Review · AnonReviewer1 · 2020-10-28
**Some part of the projects feel rushed and fail to clearly show the benefits over baseline.**

**Rating:** 4
**Confidence:** 4

**Review:**

### Summary

The paper describes a novel method to combine two different school of thoughts for improving offline reinforcement learning. The model free part relies on keeping policy changes local and model based part uses model uncertainties to ‘fine tune’ to obtain final offline RL policy.


### Strong Points

1. Paper is well written. This is impressive as both the methods on which the paper relies on are recent additions to the literature and as such I wasnot fully aware of the techniques described therein. However, despite this the paper was easy to follow and a pleasant read.
2. I would also like to thank authors for honestly describing the scenarios in which fine tuning led to degradation in the performance of the proposed method


### What can be improved

1. The paper relies on AWAC method which is itself novel and not peer reviewed.
   With benefit of doubt to authors I assume that the AWAC  represents excellent
   scholarship. This though raises another question as to how much contribution
   the results from have on their own. There maybe excellent points the authors
   use to improve upon AWC results, however with the time budget allocated to
   this review it is impossible to read and understand AWAC and then objectively
   judge the improvements bought about by this scholarship.
2. Again, I would like to commend authors for the future work directions
   mentioned in the conclusion. Unfortunately, these are the very same questions
   I would have hoped this paper addresses. For example, one of the classic use
   cases for offline RL would be that we are not aware of the situation, whether
   the data is ‘expert’ policy or ‘naive’ policy, so it is very challenging to
   decide whether or not to fine tune the results one obtains from AWAC for
   example.
3. One Page 6 penultimate paragraph: Authors say “ ... , and we are confident
   that we can learn an effective model with the available data then we do
   additional fine-tuning using MB2PO”. I am not sure how we are establish this
   ? This is a genuine question and not a rhetorical remark!
4. The details of the models used for fine tuning are absent, I did see which NN
   models are used to build dynamics models. I am sure authors would agree that
   a “neural networks that outputs a Gaussian distribution” would cover large
   literature at conference like ICLR. Moreover, what network is used would
   significantly impact what can be learned by effective referring to the
   comment above.

### Typos/ No impact on score

* Is there a ‘tilde’ missing in the algorithm 4.3, just above add sample ? if
  not do we need to learn reward model ?

* On page 6 last paragraph: The iterations mentioned are on model based samples
  (innermost For Loop in the algorithm ?)

---

> ### Author Response · Authors · 2020-11-25
> **Addressing Concerns**
>
> What can be improved
> 1. We believe that the main benefit of this work is in proposing a method that leverages both behavior-regularized model-free RL and uncertainty-aware MB RL for offline RL, and in demonstrating both theoretically and empirically the merits of such a combined approach.
>
> 2. Considering our improved results in the most recent revision, I would suggest that you use our method with alpha = 2 and omega = 0.8 in the proposed situation. Those parameters gave us our "-expert" and "-medium-expert" results. While those parameters are conservative, they were still able to achieve expert level performance on the "-expert" datasets and consistently improve on the AWAC results for the "-medium-expert" results.
>
> 3. Yes, I agree that it is hard to know for sure whether you could learn an effective model in all cases when given an arbitrary dataset. Hopefully, our newer and better results can convince you that possessing this knowledge is less required to get at least reasonable results. However, to get great results like we do on the "-mixed" datasets you probably would still need to have some idea of this in order to tune alpha and omega to be as aggressive as possible.  Still, I believe that if you have good domain knowledge of the system and the available dataset, then you could come up with a good conservative estimate for alpha and omega.
>
> 4. I added a lot of the hyperparameter details in the appendix of the most recent revision. Specifically, our dynamics models are represented by 4 hidden layer fully connected Neural Networks with 256 hidden units each and swish hidden activations. The output is a separate mean and variance for every state variable and the reward. Therefore, it can only produce diagonal gaussians.
>
> Typos
> * Yes there was a tilde missing, and we do learn the reward model as part of the dynamics model.
> * The policy iterates are each outermost for loop

---

### Official Review · AnonReviewer2 · 2020-11-03
**Benefit?**

**Rating:** 5
**Confidence:** 2

**Review:**

Hi,

First I wanted to thanks authors to put this manuscript together, I enjoyed reading it.

*Summary* Authors suggest Model-Based Behavior-Regularized Policy Optimization (MB2PO) for fine tuning policies trained with behavioral regularized model free RL.

I enjoyed reading the paper, I think it's a neat idea, and authors did a good job explaining them.
Away from the details, looking at the results I have an important concern about the usefulness of the method. So authors claim that if AWAC has not achieved expert performance. But looking at the result, the method have not delivered the promise, and in many cases the performance actually degraded. So if we cannot be (at least to a good extent) confident that fine tuning will make the performance better, what is the main benefit? Specially if we are in mostly off-policy setting, and cannot really get the performance of the model by running the policy multiple times on the environment.

I am happy to increase my score, if authors clarify my concern (/mis-understanding?).

Thanks.

---

> ### Author Response · Authors · 2020-11-25
> **New Results**
>
> With our updated results, we are now able to pretty consistently improve the AWAC results with our MB2PO fine-tuning. And even in the 4 cases were results decline, the effect is pretty negligible, and never more than 3 points.

---

### Official Review · AnonReviewer5 · 2020-11-09
**Review on the Paper "Fine-Tuning Offline Reinforcement Learning with Model-Based Policy Optimization"**

**Rating:** 4
**Confidence:** 4

**Review:**

1. Summary

This paper proposes an improved offline RL (batch RL) algorithm combining the state-of-the-art behavior-regularization actor-critic method (Nair et al., 2020) with a model-based RL technique. N-trained probabilistic dynamics models generate fictitious trajectories with uncertainty-penalized rewards after pretraining the policy with the behavior-regularization solely on the offline data. Both these generated data and the original offline data are used for the further behavior-regularized actor-critic training. Numerical results showed that the proposed method outperformed recent offline model-free and model-based RL algorithms.



2. Pros
- This paper efficiently combines the state-of-the-art method with a model-based RL technique.
- Challenges in offline RL and the related works are well-described.
- Ablation study investigated five different kinds of data types for in-depth investigation.



3. Cons

(1) Experimental Support

- The ablation study is unfair. The results of 'ours' in Table 1 are the maximum combination of AWAC and AWAC+MB2PO, which includes five best results from AWAC out of nine best results from 'Ours.' Although related explanations are written at the end of Section 5, AWAC+MB2PO should be considered as 'ours.' This is the reason why I had no choice but to give a harsh rating.
- Besides, model-based learning and generating fictitious samples suffer from compound errors, so the learning can be inaccurate, especially when the length of the trajectories is long. However, the proposed algorithm uses 95% of fictitious samples in the model-based fine-tuning phase, causing degradation of overall performance due to the compound error issues.

(2) Novelty

This work is not novel since this paper's algorithm heavily depends on state-of-the-art AWAC (Nair et al., 2020), and the uncertainty-penalized reward generalization is from other previous work (Yu et al., 2020).

(3) Reproducibility

The paper's experiments do not guarantee reproducibility since the authors did not provide the implementation code.



4. Minor concerns
- The notion of the behavioral policy should be unified, either pi_b or pi_beta.
-  The notion of the expectation should be unified either E or mathbb{E}.



=== Post Rebuttal Responses ===

I thank the authors for their replies and the corresponding new revision. After reading them and the other reviews carefully, I changed the rating accordingly. However, I still lean to reject this paper for the following reasons.

1. When saying "our method outperforms something," the standard deviation and the mean should be considered (such as confidence interval or statistical test). However, the authors seem only to consider the mean in the paper and the other replies. In this sense, the proposed method does not outperform CQL. Besides, the variances of CQL in ‘halfcheetah-mixed’ and ‘hopper-mixed’ are required to conclude that the proposed method outperforms CQL in the ‘mixed’ setting (e.g., the variance of the proposed method in 'hopper-mixed' is high).
2. I agree with the other reviewers’ common concerns on the novelty / main benefit of this paper.
3. The supplementary code can be shared with the reviewers using an anonymous link, as explained in ICLR Author Guide. It would have been better if the authors used this functionality of ‘official comments’ during the review process for better reproducibility.

---

> ### Author Response · Authors · 2020-11-25
> **Addressing Concerns**
>
> Experimental Support
>
> * We have improved our results such that we can now consistently improve over AWAC with our MB2PO. Thus, the ablation study should be much fairer now.
>
> * The fictious samples are actually collected as h-length sub-trajectories starting at real data points, where h is always 1 or 5. This should avoid the compounding error problem while still allowing us to collect some novel data. We tried to make this a bit clearer in the most recent revision
>
> (2) Novelty
> We believe the main novelty of our approaches is in demonstrating the benefits of combining both behavior-regularized model-free RL and uncertainty-aware MB RL for offline RL.
>
> (3) Reproducibility
> We included more hyperparameters in the appendix to help on this front, but also we would be happy to release our code once things are deanonymized.
>
> (4) Minor Concerns
> We believe we fixed these issues in the most recent revision.

---

### Decision · Program_Chairs · 2021-01-07
**Final Decision**

**Decision:**

Reject

**Comment:**

This paper proposes a method for offline reinforcement learning methods with model-based policy optimization where they first learn a model of the environment to learn the transition dynamics, a critic and the policy in an offline manner. They basically learn the model by training an ensemble of probabilistic dynamics models represented by neural networks that output a diagonal Gaussian distribution over the next state and reward. Then they use the covariance of the probabilistic dynamics model to get an uncertainty measure that they incorporate into the  reward when training it with the AWAC.

There were two main concerns raised by the reviewers:

1) Experiments: As pointed out by the reviewers, the experimental gains don't look very convincing. In particular, the performance of AWAC looks bad, and MB2PO doesn't give much gains on top of it. It is not clear how much better the proposed method is doing on the tasks that it does well, without any confidence intervals or variance measures provided.

2) Novelty:  This is almost a trivial combination of two existing ideas: model based policy optimization and AWAC. It is not clear how useful this particular combination is in practice, and it seems like there is not much insights gained from it.

I think better motivations, further ablations and more empirical analysis to understand the proposed model better. For example, analyzing the types of behaviors learned or how calibrated the uncertainty estimates that is incorporated into the reward is or some hyperparameter sensitivity analysis would make the paper more interesting.

As it stands right now, I am suggesting to reject this paper. I hope the authors will improve the paper for the future...